# Tools for the Assessment of Risk-Taking Behavior in Older Adults with Mild Dementia: A Cross-Sectional Clinical Study

**DOI:** 10.3390/brainsci13060967

**Published:** 2023-06-19

**Authors:** Charline Compagne, Damien Gabriel, Lénaïc Ferrero, Eloi Magnin, Thomas Tannou

**Affiliations:** 1UR LINC, Université de Franche-Comté, 25 000 Besançon, France; 2CIC-1431 INSERM, Centre Hospitalier Universitaire, 25 000 Besançon, France; 3Plateforme de Neuroimagerie Fonctionnelle Neuraxess, 25 000 Besançon, France; 4CHU Département de Neurologie, Centre Hospitalier Universitaire, 25 000 Besançon, France; 5CIUSS Centre-Sud de l’Ile-de-Montréal, Centre de Recherche de l’Institut Universitaire de Gériatrie de Montréal, Montréal, QC H3W 1W5, Canada

**Keywords:** mild cognitive impairment, Alzheimer’s disease, risky decision-making, clinical assessment, behavioral tasks, neuropsychological assessment

## Abstract

Diseases such as Alzheimer’s cause an alteration of cognitive functions, which can lead to increased daily risk-taking in older adults living at home. The assessment of decision-making abilities is primarily based on clinicians’ global analysis. Usual neuropsychological tests such as the MoCA (Montreal Cognitive Assessment) cover most of the cognitive domains and include mental flexibility tasks. Specific behavioral tasks for risk-taking, such as the Balloon Analogue Risk Task (BART) or the Iowa Gambling Task (IGT), have been developed to assess risk-taking behavior, particularly in the field of addictology. Our cross-sectional study aims to determine whether the MoCA global cognitive assessment could be used as a substitute for behavioral tasks in the assessment of risky behavior. In the current study, 24 patients (age: 82.1 ± 5.9) diagnosed with mild dementia completed the cognitive assessment (MoCA and executive function assessment) and two behavioral risk-taking tasks (BART, simplified version of the IGT). Results revealed no relationship between scores obtained in the MoCA and behavioral decision-making tasks. However, the two tasks assessing risk-taking behavior resulted in concordant risk profiles. In addition, patients with a high risk-taking behavior profile on the BART had better Trail Making Test (TMT) scores and thus retained mental flexibility. These findings suggest that MoCA scores are not representative of risk-taking behavioral inclinations. Thus, additional clinical tests should be used to assess risk-taking behavior in geriatric settings. Executive function measures, such as the TMT, and behavioral laboratory measures, such as the BART, are recommended for this purpose.

## 1. Introduction

Aging can be associated with cognitive impairments such as Alzheimer’s disease and related disorders, which affect the quality of life of older adults [1]. These progressive neurodegenerative diseases cause alterations in cognitive functions, among which are executive functions, limbic circuits and memory skills [2,3]. Functional alterations resulting from these disorders vary according to the type and duration of the causal pathology. One common feature of cognitive disorders is that they lead to a loss of functional independence and alter self-determination abilities (decision-making) [4,5]. Decision-making is a complex phenomenon involving several cognitive and behavioral components. Indeed, it requires executive functions such as planning, mental flexibility, inhibition skills and working memory [6], but also long-term memory and emotional processing [7]. The decision-making process mobilizes these skills to evaluate the consequences of possible alternatives, with the objective of making the most advantageous decision. Making an appropriate decision implicates being aware of the advantages and disadvantages of a choice, as well as personal limitations. 

Risk-taking behaviors are defined by risky conduct that can put people in danger, involving motivation and impulsivity [8]. Risky decision-making abilities represent the functions necessary for risky decision-making such as executive functions including inhibition and working memory, as well as certain cognitive functions such as attention. Indeed, the evolution of neurocognitive diseases is associated with an increase in the level of daily risk-taking at home (i.e., risk of falling, forgetting to turn off the gas, eating expired food) with consequences for older adults when the risks materialize [9,10]. Assessing older patients’ decision-making abilities and risk-taking behaviors is, therefore, an important step for the reduction of their risks in daily life activities.

To date, the assessment of cognitive and decision-making abilities, and risk-taking behavior is primarily based on a standardized geriatric assessment. It includes a clinical evaluation, neuropsychological testing, and an interview with the older adult about their life project and decisions concerning their future. To assess risky decision-making abilities, physicians usually perform global assessments of patient’s cognitive functions and daily living skills, in order to screen for the presence of major neurocognitive disorders [11]. According to the DSM-V [12], a major neurocognitive disorder (also known as dementia) is defined by impairment of at least two cognitive functions assessed by tests, including memory, gnostic, language, praxis and executive functions, that impact daily living activities. Cognitive impairment develops slowly, representing a change compared to the patient’s previous abilities, and is not explained by depressive symptoms.

The Montreal Cognitive Assessment (MoCA) is one of the most widely used instruments for the global assessment of cognitive functions and decision-making in geriatric medicine [13,14]. The MoCA is a screening instrument used for the detection of cognitive impairment, from the stage of mild cognitive impairment (MCI) to severe dementia. It assesses different aspects of cognition such as visual-spatial abilities, memory, attention, language, and orientation in time and space. Thus, this instrument also provides measures of executive functions. Mini-Mental Status Examination (MMSE) [15] and MoCA scores contribute to the evaluation of the dementia stage (mild, moderate, or severe cognitive impairment). The MoCA seems to be more predictive of cognitive impairment in the early stages than the MMSE [16], which justifies its preferential use as a global assessment tool in first evaluations. However, performance is influenced by sociodemographic variables such as age, education, and sex [17,18,19,20]. 

Thus, additional tests that specifically assess executive functions are sometimes used: the Frontal Assessment Battery (FAB) [21], and the Stroop Color-Word Test-Victoria (Stroop) are used for the assessment of inhibition [22] while the Trail Making Test assessment (TMT) [23], provides a measurement of mental flexibility. Although these instruments may help to complete the clinical profile, they are not systematically used, except by neuropsychologists.

In addition to these tests, specialized tasks have been designed to assess complex behavior abilities but are currently used primarily in research. Some of these tasks explore decision-making behavior by modeling choices involving monetary gain or loss such as the Balloon Analogue Risk Task (BART) and the Iowa Gambling Task (IGT). The BART [24] is a computerized behavioral risk-taking and decision-making task where the objective is to obtain the highest score by virtually inflating a series of balloons that may explode. In IGT [25], the aim is to maximize the final score by making a series of advantageous and disadvantageous choices of cards. Currently, these two tasks are used to assess impaired risk behavior in various medical specialties, such as psychiatry or addictology. The BART and IGT performances have been found to be correlated with real-life risk-taking, such as tobacco use, gambling, drugs and alcohol use [26,27,28,29]. In the BART, previous studies demonstrated that healthy older adults had significantly higher risk aversion and lower BART performance than younger adults [30,31,32]. In addition, older adults performed worse in the IGT (i.e., winning a smaller amount of money and doing more disadvantageous choices) compared to younger adults, which is indicative of difficulties in the appreciation of advantages in decision-making under ambiguous conditions [33,34]. These decision-making tasks provide information on the functioning of executive functions and attention, in particular with regard to reasoning while implementing a strategy, as well as on memory and mental flexibility, as they require behavioral adaptations that vary depending on the progress of the task [35,36,37,38]. The combination of these parameters with the incertitude of task conditions in the BART and the IGT contribute to the assessment of global risk-taking behavior, an important element that may eventually support clinical diagnosis.

The MoCA scale is the most common assessment used in geriatric practice to screen for impairments in cognitive function. Decision-making abilities and risk behavior capacities are often deduced through this test and clinical interviews conducted by physicians. Nevertheless, we hypothesize that the global assessment of cognitive functions using the MoCA is not specific enough to assess decision-making abilities and risk-taking behavior in cognitively impaired older adults.

The main goal of our study was thus to determine if MoCA performance correlates with the BART and IGT experimental tasks with regard to the assessment of risky decision-making behavior. The second purpose was to assess the relationship between neuropsychological measures of executive function and risk-taking behavioral data from BART and IGT.

## 2. Materials and Methods

### 2.1. Study Design and Data Sources

Our study was a cross-sectional French retrospective study conducted in the geriatric memory clinic of one French University Hospital Center. We used a PECO framework (population, exposure, comparator, and outcomes) to develop our research question [39]. The study-focused question was: “Is the clinical MoCA score (E) in patients with mild dementia (P) sufficient to measure risk-taking behavior (C) compared with experimental behavioral tasks (O)?”
P—Patients with mild dementiaE—MoCA scoreC—Assessment of risk-taking behavior (IGT net scores and BART adjusted average number of pumps)O—Experimental behavioral tasks (BART and IGT)

We collected data from electronic medical and neuropsychological records immediately after participants took part in the experimental testing, to avoid any influence on results. The data categories selected were chosen from risky decision-making studies in the older population. The present study is reported according to the Strengthening the Reporting of Observational Studies in Epidemiology (STROBE) statement guidelines [40].

### 2.2. Population

The inclusion criteria for patients in the study were as follows: (i) right-handed and over 70 years of age, (ii) with a diagnosis of mild Alzheimer’s type dementia recently diagnosed (i.e., 6 months to 2 years) by a geriatrician, (iii) have completed two behavioral computerized tests assessing risk-taking behavior (BART and IGT), (iv) have benefited from a battery of neuropsychological tests focused on executive functions, (v) demographic and neuropsychological data extracted from medical records had to be complete. These tests and the analysis of scores are described below. 

The clinical diagnosis of mild dementia was based on the following criteria, which were attested after a clinical interview conducted by a trained geriatrician: (i) MoCA score between 18 and 25, (ii) no severe depressive symptoms on the Geriatric Depression Scale (GDS-30) (GDS-30 score lower than 20) [41], (iii) identification of an impact of cognitive impairment on independency in daily life, with a loss of at least 1 point in one of the autonomy scales: Activities of Daily Living (ADL) [42] and/or Instrumental Activities of Daily Living (iADL) [43].

The exclusion criteria for the study were as follows: (i) diagnosis of a higher stage of dementia (moderate to severe), (ii) severe depressive symptoms, (iii) incomplete clinical, neuropsychological and behavioral data.

All assessments were performed between 1 March 2021, and 30 September 2021. To that end, patients were admitted to a one-day clinic or short hospital stays for clinical and cognitive assessments, including decision-making skills evaluation.

### 2.3. Neuropsychological Measures

#### 2.3.1. Neuropsychological Assessment

The evaluation of executive functions was based on patients’ global performance, assessed by the Frontal Efficiency Battery (FAB), the Trail Making Test A and B (TMT) for mental flexibility, and the Stroop–Victoria test (STROOP) for inhibition capacities.

The FAB test evaluates executive function. It consists of six items: conceptualization, mental flexibility, motor programming, sensitivity to interference, inhibition and behavioral autonomy. The expected FAB score is at least 16.

The standard TMT consists of two parts: in Trails A, subjects connect a series of 25 numbered circles in ascending order, and in Trails B, they connect 25 circles alternating between ascending numbers and letters. The duration of the TMT (B-A) and the number of errors (B-A) were the variables used to assess mental flexibility. 

The STROOP scale contains three conditions. The first condition shows a colored board (C) where the objective is to name the colors. The second displays a board (W) on which words are written in different colors. The instruction given for this second condition is to name the color of the words while ignoring their meaning. The last condition is an interference board (I) where color words are written in different colors. The objective is to name the color of the letters of the words shown on the board, even though the color of the letters does not correspond to the meaning of the word. In the three conditions, the patient had to name the color as quickly as possible. The completion time and the number of errors were the variables used to assess inhibition capacities.

The overall score, the score per part and per test, and the completion time for timed tests were extracted from the medical data.

#### 2.3.2. Experimental Behavioral Task of Decision-Making

Decision-making scores were determined based on patient’s performance on the computerized version of the Balloon analog risk task (BART) and the Iowa Gambling Task (IGT), in a version adapted to older adults (See Appendix A for the description of the procedure). To allow progressive analyses, the BART performance was divided into four blocks, each representing 25% of the task (20 balloons), and the IGT performance was divided into five blocks, each representing 20% of the task (40 trials).

The extracted data included the adjusted number of pumps, the number of explosions and the pot’s amount for the BART. The ratio of advantageous to disadvantageous choices and the pot’s amount were the data extracted for the IGT. This allowed for the calculation of total scores and risk-taking indexes, as described in the Section 2.4.

### 2.4. Statistical Analysis

Statistical analyses were performed using Jamovi software (2.2.5) and JASP (0.17.1). The normality of each dataset was evaluated with the Shapiro–Wilk test. The level of significance was set at *p* < 0.05. To identify the presence of a link between the neuropsychological measures and risk-taking behaviors, simple linear regressions were employed. Based on the literature, factors such as age, sex, depressive symptoms, and cognitive function alterations (i.e., FAB score) were found to influence risk-taking behaviors. Similarly, age, depressive symptoms, and cognitive function alterations (i.e., FAB score) were found to influence neuropsychological test scores [44,45,46,47,48]. In order to control for these predictive factors in risk-taking (BART and IGT scores) and neuropsychological assessments, a multiple regression linear analysis model was used. The impact of each predictor was tested.

#### 2.4.1. Risky Behavior: BART and IGT

Risky behavior scores were calculated for each task. In BART, scores were computed using the adjusted average number of pumps (i.e., the average number of pumps in trials ending with collection), the number of balloon explosions and the reward amount. In IGT, the relative risk was determined by calculating the “net scores” (i.e., the proportion of disadvantageous choices subtracted from advantageous choices), and the reward amount. 

The learning effect was measured for each task using repeated measures ANOVA with the number of adjusted average pumps for the BART task and the net scores for the IGT task. Then, to assess the risk-learning behavior, a Student’s paired *t*-test was performed to compare the performance at the beginning and the end of the two tasks. In the BART, the number of adjusted average pumps in the first block was compared to the last three blocks. In the IGT, the net scores at the first 80 trials was compared to the last 80 trials. Then, a simple linear regression was used to identify the presence of a link between BART (i.e., the adjusted number of pumps, number of balloon explosions and the reward amount) and IGT (i.e., net scores and reward amount) performances.

#### 2.4.2. Choices Behaviors in IGT: Rigidity, Flexibility, Win-Stay, Win-Shift, and Lose-Shift Scores

IGT allows specific analyses of participants’ strategies. Indeed, to identify changes in the strategy and behavioral adaptations of patients during the IGT, the rigidity, flexibility, win-stay, win-shift and lose-shift scores were collected. The rigidity score is the percentage obtained for the preferred choice strategy (higher-risk or lower-risk choice). The flexibility score corresponds to the proportion of switches between advantageous and disadvantageous choices. Win-stay scores represent the choice made for the same option after receiving a reward, whereas win-shift scores correspond to the proportion of changes after positive feedback. Lose-shift scores, which reflect the aversion to a negative outcome, were assessed by calculating the proportion of switches after a penalty outcome. 

Behavioral measures (rigidity, flexibility, win-stay, win-shift, lose-shift) were calculated, in accordance with previous studies on the IGT [49,50], using Student’s paired *t*-tests to compare trials at the beginning (i.e., first 40% of the task, which represent 80 first trials) and at the end of experimental tasks (i.e., last 40% of the task, representing last 80 trials).

#### 2.4.3. Relationship between Neuropsychological Scales and Experimental Tasks

To assess possible links between neuropsychological measures and risky behavioral patterns at the BART and the IGT, simple linear regression was used for comparisons. 

First, as it was our main objective, the link between MoCA scores and BART (average adjusted number of pumps, number of explosions, reward amount) or IGT data (net scores, reward amount) were calculated to find associations between the assessment of cognitive dysfunctions and risky behavior. 

Scores in assessments of executive functions [FAB (global score), TMT (completion time and the number of errors), STROOP (completion time and the number of errors)] were also matched to risky behavior evaluated by BART and IGT, to identify if BART or IGT performances were influenced by the severity of the executive function impairment. 

### 2.5. Bias Reduction Measures

To reduce the selection bias, an immediate retrospective design was used in this study for all eligible patients hospitalized during the recruitment period. Several trained geriatricians were involved in the selection process, and data analysis was performed by an independent clinician-analyst in a blinded process.

### 2.6. Ethical Requirement

Data were anonymized. Patients included in the cohort and alive at the time of data analysis were informed of the study objectives and were given the possibility to ask for their personal information to be removed. As the data were anonymous and derived from routine clinical practice, prior ethics board approval was not required. However, the use of the data for research purposes was declared by the promoting institution to the Commission Nationale de l’Informatique et des Libertés (CNIL) in France, under methodology MR004, with internal reference 2023/752. Participants did not receive compensation for their participation in the study, since all tests were already included in their care pathway.

## 3. Results

### 3.1. Demographic Data and Clinical Measures

Data from twenty-seven older patients were potentially eligible to be analyzed in the current study. However, one patient dataset was excluded because of incomplete data, and two patients’ data were withdrawn due to motor fatigue (the examiner had to press the IGT/BART response box under the patient’s direction).

Data from a total of 24 older patients (14 females) were included in the study. Their mean age was 82.1 ± 5.9 years (range: 72–97). All patients were right-handed and had mild dementia with a mean MoCA score of 21.4 ± 2.4 (range: 18–25) and a mean GDS-30 of 7.7 ± 4.4 (range: 2–18).

Patients’ demographics are displayed in Table 1. Clinical characteristics and patients’ cognitive scores are presented in Table 2. The main feature concerning neuropsychological evaluation is the fact that all patients had executive dysfunctions, their main impairments being associated with inhibition and mental flexibility. 

### 3.2. Behavioral Tasks Performances (BART and IGT)

Concerning behavioral tasks, BART results are described in Table 3, and IGT performances in Table 4. In the BART, an effect of the block on the average adjusted number of pumps was found in a first analysis (*p* < 0.05) with repeated measures ANOVA, but post hoc analyses showed that this effect was not significant. A learning effect was revealed with Student’s paired *t*-test since the average adjusted pumps in block 1 were significantly lower than the average adjusted pumps over the last three blocks (*p* < 0.05). 

For the IGT, although the effect of the block on the net scores was analyzed, no significant effect was found with repeated measures ANOVA. Additionally, Student’s paired *t*-test did not reveal a significant difference between the average of net scores for the first 80 trials and the last 80 trials. 

For the IGT, Student’s paired *t*-test showed that the win-stay choices were significantly lower at the beginning than at the endpoint performance of the task (r = −3.267, *p* < 0.01). The lose-shift choices (r = 2.645, *p* < 0.01) and the flexibility scores (r = 2.722, *p* < 0.01) were significantly higher at the beginning than at the end of the task. The evolution of choices during the IGT is displayed in Figure 1. No significant difference was found for the rigidity scores and the win-shift choices between the beginning of the task and the endpoint performance.

### 3.3. Relationship between the Experimental Tasks: BART and IGT

First, a multiple linear regression showed that the age and the sex of patients, the depressive symptoms, and the FAB did not predict the BART average adjusted number of pumps (F (4.18) = 0.86, r = 0.402, *p* = ns), the IGT net scores (F (4.18) = 0.03, r = 0.09, *p* = ns), and the IGT reward (F (4.18) = 0.54, r = 0.329, *p* = ns). A simple linear regression showed a link between the average adjusted number of pumps of the BART and the IGT net scores (r = 0.179, *p* < 0.05) (Figure 2). Additionally, a link existed between the average adjusted number of pumps at the BART and the amount of IGT reward (r = 0.494, *p* < 0.01). 

In brief, these data revealed that patients who took more risks, (i.e., who had a high number of adjusted pumps), made more disadvantageous choices on the IGT leading to a low net scores and reward amount. These results indicate consistency between risk-taking behaviors in the two tasks: BART and IGT.

### 3.4. Relationship between the Montréal Cognitive Assessment (MoCA) and the Experimental Tasks of Decision-Making

First, a multiple linear regression analysis showed that the age of patients, the depressive symptoms, and the FAB score did not predict the MoCA scores (F (3.19) = 1.76, r = 0.467, *p* = ns), the average adjusted number of pumps at the BART (F (3.19) = 1.76, r = 0.395, *p* = ns), the amount of BART reward (F (3.19) = 1.80, r= 0.471, *p* = ns), the IGT net scores (F (3.19) = 0.05, r = 0.090, *p* = ns), and the amount of IGT reward (F (3.19) = 0.48, r = 0.267, *p* = ns). Second, by using a simple linear regression, there was no significant relation between MoCA scores and the BART average adjusted number of pumps (r = 0.282, *p* = ns), MoCA scores and the BART reward (r = 0.291, *p* = ns), MoCA scores and the IGT net scores (r = 0.064, *p* = ns), and the IGT reward (r = 0.110, *p* = ns). As such, our data showed that there was no relationship between the MoCA scores and the risk behavior as it was evaluated by our two experimental tasks.

### 3.5. Relationship between Neuropsychological Measures and Experimental Task

#### 3.5.1. Neuropsychological Measures and BART

First, a multiple linear regression demonstrated that the age of patients, the depressive symptoms, and the FAB score did not predict significantly the score at TMT completion time (F (3.19) = 0.88, r = 0.351, *p* = ns) and TMT errors (F (3.19) = 0.49, r = 0.268, *p* = ns). A simple linear regression showed a link between the number of BART explosions and TMT completion time (B-A) (F (1.22) = 7.20, r = 0.497, *p* < 0.01) as well as with the number of total TMT errors (F (1.22) = 7.04, r = 0.492, *p* < 0.01) (Figure 3).

Similarly, a multiple linear regression demonstrated that the age of patients, the depressive symptoms, and the FAB scores did not predict significantly the score at the B part of TMT completion time (F (3.19) = 1.03, r = 0.375, *p* = ns) and errors (F (3.19) = 0.52, r = 0.276, *p* = ns). A simple linear regression showed a significantly negative link between the number of BART explosions and the completion time (part B) (F (1.22) = 7.10, r = 0.494, *p* < 0.01), and also with the number of errors (F (1.22) = 6.80, r = 0.486, *p* < 0.01).

No relation was found between BART variables of risky behavior and the Stroop scale.

#### 3.5.2. Neuropsychological Measures and the IGT

No significant association was found between IGT scores and neuropsychological measures (Stroop, TMT).

## 4. Discussion

Our study investigated whether the risky decision-making behavior, assessed with experimental tasks such as the IGT and the BART, correlates with MoCA scores in older patients with mild dementia. Based on behavioral data, no relationship was found between MoCA scores and those from behavioral decision-making tasks. As such, the lack of relationship between MoCA and these experimental tasks seems to show that, as a global assessment of cognitive functions, the MoCA is not specific enough to assess risky decision-making processes. Moreover, we found that patients with riskier behavior in the BART (high average adjusted number of pumps), also made riskier choices in the IGT (low net scores and reward amount). These results suggest that, unlike the MoCA, these two experimental tasks specifically evaluated the same risk-taking decision-making process. Additionally, a relationship was found between risky behavior in the BART and mental flexibility, as evaluated by the TMT. Given the additional association between the BART and the TMT, and the fact that patients reported enjoying more the BART than the IGT, the BART appears to be a promising task for the clinical assessment of decision-making capacity in older adults. 

### 4.1. Implication for Clinical Practice

The MoCA is widely used in clinical practice since it assesses most cognitive functions, including executive functions. In clinical follow-ups, the MoCA targets potential dysfunctions of the main cognitive abilities needed for independent living. That is why it is so useful for physicians. However, even though it is sometimes used as a proxy for general cognitive evaluation, our study demonstrates that it does not provide accurate information on decision-making abilities that may impact daily life skills. Thus, objectifying risk-taking behaviors is a key issue in supporting the aging-in-place of older persons living with dementia. To evaluate the decision-making process, experimental tasks such as the BART and the IGT can be used as complementary tools to explore the risky decision-making behavior of participants.

Another interesting point of our study was the demonstration of a relationship between risky behavior in the BART and mental flexibility, as evaluated by the TMT. We observed that patients with successful high-risk behavior (high adjusted pumps and number of explosions) achieved faster completion times on Trail B and Trail B-A, and had a lower error rate on trail (B-A). TMT is more commonly used in routine practice. Although it is not a test routinely performed in geriatric consultations, it is often used in standard neuropsychological assessment, and for the screening of driving abilities. Specifically, completion time on the TMT is used to assess executive functions, in particular mental flexibility, and processing speed. Previous studies have shown that performance on the TMT varied with age and education [23], especially with regard to processing speed [51]. Indeed, older adults had poor performance on the TMT, as their number of errors was higher and their completion time was longer [23,52,53]. 

Thus, our results illustrate that patients who scored higher on the BART had better TMT scores, which suggests that they retained more mental flexibility. Some authors argue that a highly explosive behavioral profile at BART (and IGT) may be explained by impulsivity [54,55,56,57]. 

Previous studies suggested that risky behavior in behavioral tasks might be explained by some health conditions. Patients with mild cognitive impairment made more disadvantageous decisions [22,58,59] or took more risks in the hope of a higher gain on the IGT [60] compared to older adults in the control group. Disadvantageous decision-making profiles could be related to a higher level of apathy [4]. Another study demonstrated that patients in the early stages of Alzheimer’s disease had not established a risk strategy during the IGT, and frequently switched between safe and risky choices [61].

However, our data may suggest that some high risk-taking behavioral profiles (higher number of explosions and average adjusted number of pumps) can be supported by higher mental flexibility abilities, as demonstrated by short TMT completion time and fewer errors, in comparison with low older adults with BART risk-taking profiles. Patients with a “pathological” TMT score (long completion time and high number of errors) made fewer risky choices on the BART (low number of explosions and adjusted number of pumps). This could be explained by an increase in conservative choices and high-risk aversion in older adults [62]. However, risk aversion was associated with poorer decision-making in older adults. Previous studies also demonstrated that risk-taking in older adults was lower than in younger adults [63,64]. 

Based on our data, and pending further and larger studies, it might be suggested to physicians, in cases of patients with cognitive impairments, to investigate decision-making skills with more specific neuropsychological tasks, such as the TMT. If uncertainty persists, more specific behavioral tasks, such as the BART or the IGT, could be useful. Among these behavioral tasks, BART seems to be preferable, given its good acceptability in older adults and its link with mental flexibility tasks.

### 4.2. Implication for Future Research

The present study demonstrated the feasibility of behavioral and neuropsychological tasks assessing risk-taking in older adults with mild dementia. Indeed, few studies have addressed this issue. Our study is one of the first to involve this type of data collected from an older clinical population.

In experimental settings, the decision-making process also comprises sub-risk (known consequences of choices) and sub-ambiguity (unknown consequences) decisions. However, the repetition of tasks, even under ambiguity, leads to the learning of the rule and subrisk decisions. The confounding effect of learning can be analyzed by successive score blocks. In our study, behavioral results demonstrated a learning process during the task, both at the BART and the IGT. According to previous studies, the average adjusted number of pumps in the BART was significantly lower in the first block, compared to the last three blocks [56,65]. A previous study demonstrated that there may be a progressive shift from an under-uncertainty decision-making condition at the beginning of the task to an under-risk decision-making condition during the BART [66]. While no effect of the block on the net scores was found in the IGT, the behavioral data changed between the beginning and the end of the task. In the last trials, patients tended to make the same choice after a gain, and switched choices less frequently after a loss, in comparison with the first trials [49]. In the two tasks, the behavioral data demonstrated a change of strategy between the beginning and the end of the task. The difference observed between the first and the last trials could also have been influenced by a learning or practice effect [67]. Thus, the first trials of both tasks appear to be more associated with exploratory behavior than with risk-taking behavior. 

### 4.3. Limitations and Future Research

Of course, despite its clinical contribution, our study has some limitations. First, its impact is limited by a relatively small sample size, which could affect the significance of observed links. Second, the computerized BART and IGT tasks assess decision-making in an experimental setting, and not in real conditions. This type of experimental task, initially designed for younger patients, can be difficult and tiresome for older patients, because of their duration. Another limit to consider is that patients did not receive compensation for their participation, since all tests were included in their care pathway. However, previous studies demonstrated the nature and the magnitude of the reward impact risk-taking behavior; risk aversion considerably increases after negative feedback in conditions comprising large real monetary rewards but does not increase as much in trials with large hypothetical rewards [68,69]. Future works should also consider the impact of impulsivity and thus include an impulsivity scale such as the BIS or the UPPS in the neuropsychological tools used to compare the clinical score with behavioral results obtained in the BART and IGT. Concerning the implication for clinical practice, behavioral tasks such as the IGT or the BART can only be performed on a computer and may not be so easy to introduce in clinical practice. Indeed, their administration is more complex than usual tests, takes longer and is computerized, which explains why they are not yet routinely used in clinical settings. Consequently, the conditions for optimizing their use in the clinical pathway need to be better defined by further studies. These results are encouraging, but future studies will be needed. 

## 5. Conclusions

Behavioral results revealed no relationship between the MoCA and two risky decision-making tasks: the BART and the simplified IGT in patients with mild dementia. These results suggest that the MoCA, as a routine clinical tool, does not seem to be specific enough to assess risky behavior. Indeed, the MoCA is widely used in current practice as it allows for a global assessment of cognitive processes, including those involved in the decision-making process. However, while it is a very informative tool that targets the dysfunction of different cognitive processes, it does not provide information on the patient’s behavior in risky conditions. Thus, additional clinical tests should be used to assess risk-taking behavior in geriatrics, particularly executive function measures such as the Trail Making test and behavioral laboratory measures such as the BART. Based on our results, patients with mild dementia who performed higher on the BART also had better TMT scores, suggesting greater mental flexibility. 

Behavioral laboratory tasks may have a real interest as part of the assessment of the risk-taking behavior of patients with mild dementia. The assessment of decision-making abilities and risky behavior, as a complex phenomenon requiring different cognitive components, may be facilitated by using a combination of several tests, both clinical and behavioral. The inclusion of behavioral laboratory tasks could be particularly beneficial for older adults living alone, where there may be limited access to behavioral data in their daily living contexts.

## Figures and Tables

**Figure 1 brainsci-13-00967-f001:**
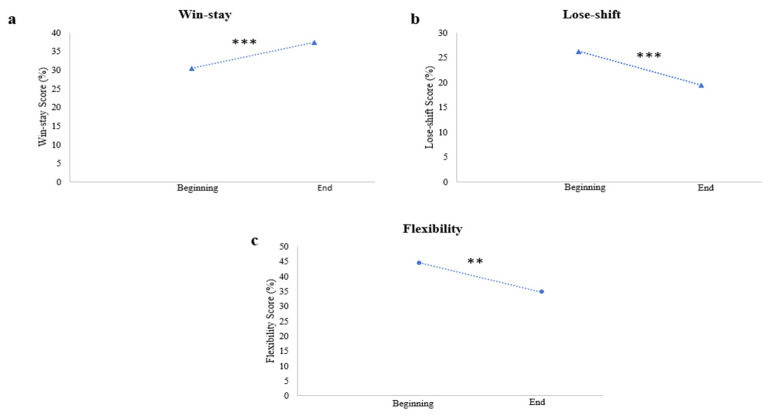
Evolution of choices during the IGT. (**a**) Evolution of choices in the win-stay choices between the beginning and the end of the task. (**b**) Evolution of choices in the lose-shift choices between the beginning and the end of the task. (**c**) Evolution of choices in the flexibility scores between the beginning and the end of the task. (Significant differences were noted by using ** *p* < 0.01; *** *p* < 0.001).

**Figure 2 brainsci-13-00967-f002:**
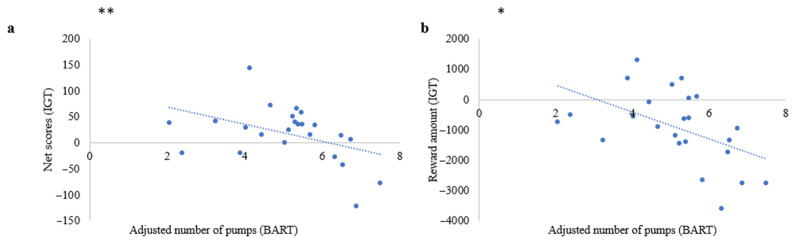
Relationship risky behavior between BART and IGT. (**a**) Association between the net scores at the IGT and the average adjusted number of pumps at the BART. (**b**) Link between the reward amount at the IGT and the average adjusted number of pumps at the BART. (Significant differences were noted by using * *p* < 0.05; ** *p* < 0.01).

**Figure 3 brainsci-13-00967-f003:**
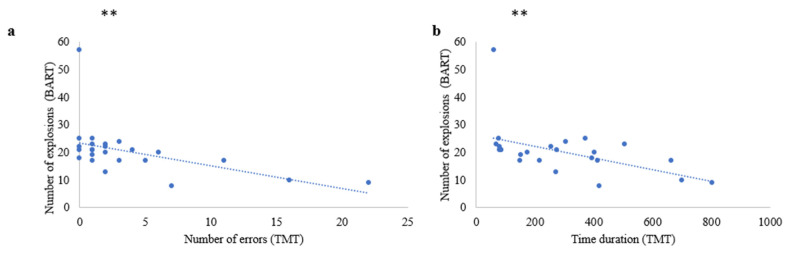
Relationship risky behavior between TMT score and BART. (**a**) Negative link between the number of explosions at the BART and the number of errors at the TMT. (**b**) Negative link between the number of explosions at the BART and the time duration at the TMT. (Significant differences were noted by using ** *p* < 0.01).

**Table 1 brainsci-13-00967-t001:** Patients demographics.

Demographic Variables				
Age	82.08 (5.95)			
Female	14			
Years of education	10 (3.38)			
Family status	Married (13)	Widowed (8)	Divorced (1)	Single (2)
Type of major neurocognitive impairment	Alzheimer (14)	Vascular (5)	Late (4)	Mixed dementia (1)
Hospitalization	Hospitalized one day (15)		Hospitalized several days (9)	

**Table 2 brainsci-13-00967-t002:** Clinical characteristics and patient’s cognitive score.

	MoCA	GDS30	ADL	iADL	FAB	STROOP (if)	STROOP (IF)	TMT (Duration)	TMT (Errors)
Mean	21.42	7.66	5.23	4.63	13.3	1.68	3.69	291.7	3.79
Median	24.5	6.5	6	4.5	14	1.65	3.13	263.29	2
SD	2.41	4.44	1.01	1.88	2.48	0.44	1.46	214.53	5.46

**Table 3 brainsci-13-00967-t003:** Descriptive statistics for all measures of the BART and per block of 20 balloons.

	All Block, M(SD)	Block 1, M(SD)	Block 2, M(SD)	Block 3, M(SD)	Block 4, M(SD)
Average adjusted pumps	5.14 (1.36)	4.88 (1.37)	5.23 (1.49)	5.50 (1.74)	5.33 (1.57)
Popped balloons	20.42 (9.13)	4.68 (1.13)	5.21 (1.79)	4.33 (1.40)	5.13 (2.31)
Reward amount	2283.33 (986.48)	540 (117.14)	526.68 (149.25)	527.92 (95.28)	550 (183.87)

**Table 4 brainsci-13-00967-t004:** Descriptive statistics for all measures of the IGT and per block of 40 trials.

	All Block, M(SD)	Block 1, M(SD)	Block 2, M(SD)	Block 3, M(SD)	Block 4, M(SD)	Block 5, M(SD)
Net scores	17.17 (52.82)	2.25 (12.68)	3.2 (10.63)	4.08 (13.25)	4.50 (14.93)	3.08 (14.86)
Reward amount	−860 (1213.80)	−254.17 (512.22)	−641.68 (693.40)	−800 (813.74)	−872.92 (1045.80)	−875 (1191.18)

## Data Availability

Original data can be shared upon reasonable request.

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
