# Peer review of "Tools for the Assessment of Risk-Taking Behavior in Older Adults with Mild Dementia: A Cross-Sectional Clinical Study"

_brainsci, 2023, doi:10.3390/brainsci13060967_

Round 1

Reviewer 1 Report (Previous Reviewer 1)

Comments and Suggestions for Authors

I carefully read the resubmitted version of the manuscript titled "Tools for the assessment of risk-taking behavior in older adults with mild dementia: a cross-sectional clinical study." I think that Authors have addressed all the comments and issues raised by the reviewers of their previous version, and that the manuscript has significantly improved in clarity and scientifical sounding. I have no further requests.

Author Response

Thank you so much for your general comments that allowed us to improve the manuscript.

Reviewer 2 Report (New Reviewer)

Comments and Suggestions for Authors

This is a well-structured article. The main question addressed by this research is the potential tools for assessing the risk-taking behaviors in older adults with mild cognitive impairment.

The introduction gives the background of this study as it briefly describes cognitive impairment associated with aging, various tools used to evaluate cognitive functions and risk-taking behaviors in geriatric populations as well as new specific tools which could assist towards this direction.

“Materials and Methods” section is descriptive enough. It refers to the design of the study, the population, neuropsychological batteries, ethical issues and statistical analyses implemented during this study.

The results are quite interesting and, to my opinion, well presented.

The discussion is well written, summarizing and discussing the main findings of the study. The existence of a paragraph summarizing its main limitations and possible targets for future research is, to my opinion, one advantage of this study too.

Furthermore, the conclusions could perhaps be written in a more detailed manner.

References are adequate in number and relative to the subject.

Comments on the Quality of English Language

English language and style are generally fine.  Minor editing of English language is required before publication.

Author Response

We thank you for your general comments that helped us improve the manuscript.

We have modified the conclusion by adding details based on our results and their relevance to the clinical field. Please see the changes on lines 467-488.

We have corrected a few grammatical errors.

This manuscript is a resubmission of an earlier submission. The following is a list of the peer review reports and author responses from that submission.

Round 1

Reviewer 1 Report

Comments and Suggestions for Authors

The manuscript titled “Tools for the assessment of risk-taking behavior in older adults with mild cognitive impairment: a clinical study” proposes a study in order to highlight the proper tools for assessing decision-making skills and in particular risk taking skills in older adults with cognitive impairments. The aim of the research is to determine whether Montreal Cognitive Assessment (MoCA) test is able to supply to specific tools, such as Balloon Analogue Risk Task (BART) or the Iowa Gambling Task (IGT), developed for assessing risky behavior within a behavioral evaluation framework. Twenty-four older patients with cognitive impairments had completed a battery of tests (MoCA, BART, IGT, GDS30, ADL, IADL, FAB, STROOP, TMT), and their scores were analyzed via Pearson’s correlations. Results showed that MoCA and behavioral decision-making tasks scores (BART and IGT) were independent. Moreover, there was a certain degree of concordance between one of the latter experimental tasks, namely BART, and neuropsychological test usually employed in large batteries, namely Trail Making Test. Authors discussed their results in light of previous literature highlighting strengths and limitations of their results and giving some hints for further research.

I carefully read the manuscript and I think it could be of interest for the readers of Brain Sciences, although there are some points to clarify before it could be ready for publication as a research article. Below there are my comments and suggestions.

I think that the present manuscript poses a relevant question, that is “are global cognitive functioning tests able to address all cognitive domains, and in particular high-level ones which are useful for everyday life for older people?” Said that, my opinion is that we should make efforts to imagine and build tools for evaluating such domains in a more ecological manner. On one hand, we found very basic neuropsychological tests with abstract contents and stimuli, and on the other experimental tasks with complex procedures but the same degree of abstractness of the first ones. To think and to set up tasks relying on everyday life situations, especially for older people, is one of the next challenges for geriatrists, gerontologists and neuropsychologists as well.

Introduction

Lines 35-36: I think you are referring to cognitive reserve. A citation would be appropriated to support the first statement.

Lines 66: Why did you write “which contrasts”? I would write that “an increased risk of falling… comes along with the decrease of risk perception…” Indeed, the two phenomena comes together in a direct association rather than being in contrast.

Materials and Methods

Lines 161 and subsequent: citations are needed for the tools which are introduced for the first time, e.g. GDS-30, ADL, IADL et cetera.

Lines 193: in the submission portal there were no supplementary materials to download.

Statistical analysis subsection – Lines 202 and subsequent: since the majority of your statistical analysis consists in several Pearson correlations, did you perform some adjustments for multiple comparisons, in order to reduce the probability of type I error?

Results

Lines 264-266: here you wrote that “All patients were right-handed and had mild cognitive impairment”, whilst in Population subsection you stated that “We screened data of all patients aged ≥ 70 years who had been diagnosed with mild to moderate dementia”. In Table 1 you labeled your sample as having a certain “Type of major neurocognitive impairment”. I am confused of what kind of neurocognitive impairment (minor or major, according to DSM V) we are talking about. Please clarify this issue, I think it’s a big one since the clinical characteristics of your sample are not clearly specified.

Table 4: Please adjust the formatting, the numbers within the table are unreadable.

Discussion

I have no comments for this section, I think it’s clear and address properly the results of the study within the literature framework.

Reviewer 2 Report

Comments and Suggestions for Authors

The manuscript entitled ‘Tools for the Assessment of risk-taking behavior in older adults with mild cognitive impairment: a clinical study’ reports a pilot study on the cognitive domain in order to screen for the presence of major neurocognitive disorders by clinicians.

The Montreal Cognitive Assessment is widely used for the detection of cognitive impairment, and to diagnose mild to moderate and severe dementia.

In this retrospective cohort study (all patients aged ≥ 70 years) the authors aimed to demonstrate the relationship between the MoCA and two risky decision-making tasks, the BART and the simplified IGT. They showed that no one provided information on the patient's behavior in risky conditions. The conclusion is that the MoCA does not seem to be specific and useful to evaluate the risky behavior by clinicians.

The experimental plan and methods are well executed and described.

The literature cited is appropriate and relevant to the study.

I suggest providing readable information about the tests, such as a reader-friendly table and figure, to better guide the reading of the manuscript.

The length of the paper is commensurate with the message.

Reviewer 3 Report

Comments and Suggestions for Authors

Thank you for giving me the opportunity to review this manuscript.

I think there are many flaws in this manuscript.

1) The title was wrong. Please change the title. First, the target population was mild to moderate dementia. Mild cognitive impairment is different from dementia. Second, I think this study was a cross-sectional study. Please describe the study design in the title, the abstract, and the method section.

2) Please attach the strobe checklist and fill in the page numbers.

3) In the abstract, the authors described "In the current study, 24 patients (age: 82.1 ± 5.9) diagnosed with a mild or major neurocognitive disorder completed the cognitive assessment (MoCA and executive function assessment) ", but I think that this study included patients with major neurocognitive disorders only. Please modify the sentences.

4) Please describe the definition of "decision-making ability", "risk-taking behaviors", "risky decision-making abilities", and "decision-making capacities" clearly. What is the differences among them and neurocognition including inhibition, cognitive flexibility, attention, executive function, and so on? 

5) Please describe the study design by using the PECO (the population, the exposure, the control and the outcomes) format in the method section.

6) The authors described that "we hypothesize than the global assessment of cognitive functions using the MoCA is not specific enough to assess risk-taking behavior in cognitively impaired older adults.", but I don't think that this hypothesis emphasized the novelty and the clinical meaningfulness of this study. 

7) Please define all predictors, potential confounders and effect modifiers. Please describe all statistical methods, including those used to control for confounding.

8) Please describe how the sample size was arrived at. Please describe how missing data was addressed.

9) The authors described that "Our study was a French retrospective cohort study", but I think this study was a cross-sectional study, because the statistical analysis plan was not based on a cohort study.

10) Please describe what was the statistical significance in this study. what kind of tests was used to assess the correlation? Pearson?

11) How many sample size was warranted to investigate the aim of this study in future studies.

Round 2

Reviewer 1 Report

Comments and Suggestions for Authors

I carefully read the revised version of the manuscript and I think that Authors properly addressed all the issues raised by the reviewers. The manuscript has improved .

Reviewer 3 Report

Comments and Suggestions for Authors

1) The title is still wrong. There have been no word/definition of "mild dementia impairment"

2) The study design is not organized. The definition of the exposure and the control is unclear. The authors showed the outcomes both in the exposure, control, and the outcomes.
